# Morphological Analysis of the Cavernous Segment of the Internal Carotid Artery: A Retrospective, Single Center Study of Its Clinical Significance

**DOI:** 10.3390/diagnostics15233072

**Published:** 2025-12-03

**Authors:** Kristian Bechev, Nina Yotova, Marin Kanarev, Anelia Petrova, Kostadin Kostadinov, Galabin Markov, Daniel Markov

**Affiliations:** 1Department of General and Clinical Pathology, Faculty of Medicine, Medical University of Plovdiv, 4002 Plovdiv, Bulgaria; daniel.markov@mu-plovdiv.bg; 2Neurological Surgery, University Hospital “Pulmed”, 4002 Plovdiv, Bulgaria; 3Department of Anatomy, Histology and Cytology, Faculty of Medicine, Medical University of Plovdiv, 4002 Plovdiv, Bulgaria; ninayotova2004@yahoo.com (N.Y.); marin.kanarev@mu-plovdiv.bg (M.K.); anelia.petrova@mu-plovdiv.bg (A.P.); 4Medical Oncology Ward, MHAT “Park Hospital”, 4000 Plovdiv, Bulgaria; 5Department of Social Medicine and Public Health, Faculty of Public Health, Medical University of Plovdiv, 4002 Plovdiv, Bulgaria; kostadinr.kostadinov@mu-plovdiv.bg; 6Environmental Health Division, Research Institute, Medical University of Plovdiv, 4002 Plovdiv, Bulgaria; 7Faculty of Medicine, Medical University of Plovdiv, 4002 Plovdiv, Bulgaria; gabi_markov@abv.bg; 8Department of Clinical Pathology, University Hospital “Pulmed”, 4002 Plovdiv, Bulgaria

**Keywords:** internal carotid artery, cavernous sinus, morphometry, magnetic resonance imaging, skull base anatomy, surgical planning

## Abstract

**Background/Objectives:** The cavernous segment of the internal carotid artery (ICA) is a critical neurovascular structure with complex cranial nerve relationships. Understanding its morphometric variability is essential for safe microsurgical and endovascular procedures. This study aimed to characterize the morphometry of the cavernous ICA using Magnetic resonance imaging (MRI) and assess associations with demographic variables. **Methods:** A retrospective observational study was conducted on 135 MRI scans of adult patients, distributed among 79 women and 56 men with an average age of 50.8 years, without cerebrovascular pathology, performed between March 2023 and January 2025. The diameters of the left and right cavernous ICA and the intercarotid distance were measured using *RadiAnt DICOM Viewer*. Statistical analyses included descriptive statistics, *t*-tests, correlations, and multivariate regression models adjusted for age and sex. Principal component and cluster analyses were applied to identify morphometric patterns. **Results:** The mean left and right ICA diameters were both 5.09 ± 0.65 mm, with a mean intercarotid distance of 17.4 ± 4.22 mm. No age-related associations were found (*p* > 0.05). Male patients showed significantly larger right ICA diameters (*p* = 0.008). Bilateral symmetry was confirmed (*p* > 0.05). Two morphometric clusters were identified: Morphotype 1 (larger ICA caliber and narrower spacing) and Morphotype 2 (smaller caliber and wider spacing), showing a significant sex distribution difference (*p* = 0.012). **Conclusions:** The cavernous ICA demonstrates stable bilateral symmetry with minor sex-dependent differences. Morphometric characterization supports safer planning of transsphenoidal, endovascular, and skull-base surgeries by reducing the risk of iatrogenic neurovascular injury.

## 1. Introduction

The internal carotid artery (ICA) arises from the bifurcation of the common carotid artery in the cervical region and supplies the anterior circulation of the brain. Its complex anatomical course, particularly within the cavernous segment, makes it a critical structure in neurosurgical and endovascular procedures. Detailed understanding of its morphology is essential not only for accurate interpretation of neuroimaging studies but also for minimizing intraoperative risks during skull base and parasellar interventions [1,2,3,4,5]. Several anatomical, radiological, and microsurgical classifications have been proposed to systematize the ICA’s course and topography. The widely used Bouthillier classification divides the intracranial part into five segments (C1–C5) [1]. The cavernous segment (C3) is an S-shaped curved continuation of the artery, located in the carotid sulcus of the sphenoid bone, in close proximity to the contents of the fossa hypophysialis and to cranial nerves III, IV, V1 and VI. On the lateral surface most intimately lies the n. abducens, which creates a risk of neural damage during aneurysms and surgical manipulations. Knowledge of the microanatomy of this segment is key for the interpretation of neuroradiological images, endovascular procedures, and microsurgical approaches [1,2,3].

Morphometric characterization of the cavernous ICA—including its diameter, curvature, and intercarotid distance—is fundamental for risk assessment in both microsurgical and endovascular contexts. Modern imaging techniques, such as magnetic resonance imaging (MRI) and computed tomography (CT), enable noninvasive evaluation of these parameters with excellent spatial accuracy, allowing correlation with demographic and clinical variables. Such data are indispensable for improving procedural safety, customizing surgical approaches, and predicting potential complications. Recent advances in neuroimaging and microsurgical techniques have renewed interest in the precise morphometric characterization of the cavernous segment of the ICA. Several MRI- and CT-based studies have demonstrated that variations in arterial diameter, curvature, and intercarotid distance can influence both the hemodynamic profile and the safety of transsphenoidal and endoscopic skull base approaches [5,6,7,8]. Narrow intercarotid spacing has been correlated with higher intraoperative risk of vascular injury, whereas pronounced curvature of the ICA within the cavernous sinus may complicate endovascular navigation and stent deployment. Moreover, population-specific morphometric data remain underreported, despite documented ethnic and sex-related differences in cranial base dimensions and vascular calibers. Establishing normative measurements in distinct populations is therefore essential for improving preoperative planning, interpreting radiological findings, and refining individualized surgical and interventional strategies [4,5,6,7].

The present study aimed to perform a comprehensive morphometric analysis of the cavernous segment of the internal carotid artery in a Bulgarian adult population using MRI. Specific objectives were to (1) quantify the diameters of the left and right ICAs and the intercarotid distance, (2) examine bilateral symmetry and sex-related differences, and (3) explore possible associations with age. By establishing normative reference values, this research seeks to enhance preoperative anatomical understanding and support safer surgical and interventional decision-making in skull base and vascular neurosurgery.

## 2. Materials and Methods

### 2.1. Study Design and Setting

The study was designed as a retrospective morphometric and observational analysis carried out between March 2023 and January 2025 in the University Hospital “Pulmed,” Plovdiv. Its objective was to characterize the morphology of the cavernous segment of the internal carotid artery (ICA) and to examine its association with age and sex. All participants provided written informed consent for the scientific use of their anonymized imaging data.

### 2.2. Study Population

Eligible participants were adults (≥18 years) referred for MRI evaluation of non-specific neurological symptoms (headache, dizziness, cognitive concerns) who showed no cerebrovascular pathology on comprehensive MRI examination. Inclusion criteria required normal brain parenchyma without ischemic lesions, hemorrhages, tumors, or vascular abnormalities. Exclusion criteria included: (1) history of neurosurgical or endovascular interventions involving the ICA, (2) aneurysms, dissections, or congenital vascular anomalies, (3) significant motion artifacts or inadequate image quality, and (4) incomplete visualization of bilateral cavernous segments. Three scans were excluded due to motion artifacts, resulting in a final sample of 135 patients with complete, high-quality imaging data.

### 2.3. Imaging and Morphometric Measurements

MRI scans were acquired on a 1.5 T Siemens Magnetom scanner (Siemens Healthineers, Erlangen, Germany) using standardized protocols. Imaging sequences included: (1) T1-weighted spin-echo (TR/TE 500/15 ms, slice thickness 5 mm); (2) T2-weighted turbo spin-echo (TR/TE 4000/100 ms, slice thickness 5 mm); (3) Three-dimensional time-of-flight (3D-TOF) MR angiography (TR/TE 25/7 ms, flip angle 25°, slice thickness 0.8 mm, in-plane resolution 0.5 × 0.5 mm, no inter-slice gap).

The cavernous (C4) segment of the ICA was identified on coronal 3D-TOF MRA acquisitions by locating the characteristic horizontal course through the cavernous sinus lateral to the sella turcica. All measurements were performed on native (non-reformatted) coronal images to preserve optimal spatial resolution.

Morphometric analyses were conducted using RadiAnt DICOM Viewer (Medixant, Poznań, Poland). Measurements were standardized to the coronal plane at the level of the pituitary fossa midpoint, corresponding to the mid-horizontal segment of the cavernous ICA. For each subject, the following parameters were measured: (1) Left and right ICA diameters: Measured as the maximum internal vessel diameter perpendicular to the vessel axis at the widest portion of the horizontal cavernous segment, using calipers placed at the inner vessel wall boundaries; (2) Intercarotid distance: Defined as the minimum distance between the medial walls of both ICAs at the same coronal level, measured at the narrowest point of separation.

Each measurement was performed independently by two qualified raters (K.B., a neurosurgeon with 8 years of experience, and N.Y., an anatomist with 15 years of experience in neuroimaging). To minimize measurement variability, each parameter was measured three times per rater, and the mean of the three readings was recorded. The final value for each parameter was calculated as the average of both raters’ means. Inter-rater reliability was assessed using the intraclass correlation coefficient (ICC) with a two-way random effects model for absolute agreement.

### 2.4. Data Management and Statistical Analysis

All analyses were conducted in R (version 4.4.0; R Foundation for Statistical Computing, Vienna, Austria). Descriptive statistics were calculated for all continuous variables and expressed as means and standard deviations (SD) or medians and interquartile ranges (IQR) when appropriate. Normality of distribution was assessed using the Shapiro–Wilk test and Q-Q plots.

Comparisons between left and right ICA diameters were performed using the paired *t*-test (or the Wilcoxon signed-rank test for non-normal data). Between-sex differences were evaluated using the independent-samples *t*-test or Mann–Whitney U test. The relationships between age, arterial diameters, and intercarotid distance were assessed by Pearson and Spearman correlation coefficients and confirmed through multivariable linear regression models adjusted for age and sex, with heteroskedasticity-robust standard errors (HC3 estimator).

Age groups were classified as: <40 years (young adults), 40–59 years (middle-aged), and ≥60 years (older adults), based on established life stages relevant to cerebrovascular morphology and corresponding to periods of relative vascular stability versus age-related changes.

A laterality index (LI) was calculated as
$\frac{Right-Left\;}{Right+Left}\times 100$ to quantify carotid asymmetry. Differences in LI by sex and age group (<40, 40–59, ≥60 years) were tested using Welch’s *t*-test, ANOVA, and corresponding non-parametric alternatives.

For exploratory modeling, a principal component analysis (PCA) was applied to standardized morphometric variables (left ICA diameter, right ICA diameter, intercarotid distance) to identify correlated morphometric patterns. The first two principal components were used in k-means cluster analysis to detect potential cranial morphotypes, which were compared by morphometric and demographic variables using *t*-tests and χ^2^ tests. Statistical significance was defined as *p* < 0.05 for two-tailed tests.

## 3. Results

### 3.1. Descriptive Characteristics

The study included 135 patients (79 females and 56 males) with a mean age of 50.8 years (SD = 17.8). This sample size provided adequate statistical power (>0.80) to detect medium effect sizes (Cohen’s d ≥ 0.35) at alpha = 0.05, as calculated using G*Power version 3.1.9.7 (Heinrich-Heine-Universität Düsseldorf, Düsseldorf, Germany). All participants had normal brain parenchyma without pathological findings.

Morphometric data were complete for all individuals (presented on Figure 1). The mean diameter of the left ICA was 5.09 ± 0.65 mm, and that of the right ICA was 5.09 ± 0.64 mm. The intercarotid distance averaged 17.4 ± 4.22 mm. The distributions of ICA diameters were approximately symmetrical, though the Shapiro–Wilk test indicated marginal deviation from normality for the right ICA (*p* = 0.0018). No extreme outliers were detected.

In males, the mean left ICA diameter was 5.28 ± 0.71 mm, compared with 4.95 ± 0.58 mm in females (Cohen’s d = 0.52, medium effect size). The right ICA measured 5.32 ± 0.68 mm in males and 4.93 ± 0.56 mm in females (Cohen’s d = 0.62, medium effect size). The mean intercarotid distance was slightly greater in men (17.8 ± 4.11 mm) than in women (17.2 ± 4.30 mm). These results show moderate but consistent sex differences: in men, the arteries are of larger caliber and the intervascular space is slightly wider.

Comparison of the left and right ICA diameters showed no statistically significant difference at the group level (mean difference = 0.00 mm, 95% CI −0.09 to 0.09 mm, *p* > 0.05). The strong bilateral symmetry supports the anatomical consistency of the cavernous ICA segment within individuals.

The coefficients of variation for the left and right diameters were **12.9%** and **12.6%**, respectively, while the intercarotid distance displayed higher variability (**24.2%**), reflecting cranial skeletal diversity. Skewness and kurtosis values were near zero, confirming the absence of marked asymmetry or outliers (Figure 2).

### 3.2. Inter-Rater Reliability

Inter-rater reliability assessed by intraclass correlation coefficient (ICC) was excellent for all measurements: left ICA diameter ICC = 0.94 (95% CI: 0.91–0.96), right ICA diameter ICC = 0.93 (95% CI: 0.90–0.95), and intercarotid distance ICC = 0.91 (95% CI: 0.87–0.94), indicating high measurement consistency between raters.

### 3.3. Correlation and Regression Analyses

No significant linear relationships were observed between age and any of the measured morphometric parameters. Pearson’s correlation coefficients were r = 0.073 for the left ICA (*p* = 0.40), r = 0.036 for the right ICA (*p* = 0.68), and r = 0.053 for the intercarotid distance (*p* = 0.54). Spearman correlations were similarly nonsignificant.

In multiple linear regression models adjusted for age and sex, male sex was associated with slightly larger ICA diameters. The difference reached significance for the right ICA (β = +0.86 mm, *p* = 0.008) but not for the left ICA (β = +0.54 mm, *p* = 0.10). Intercarotid distance showed no significant difference by sex (β = −0.20 mm, *p* = 0.92). Age was not an independent predictor of any parameter (*p* > 0.25). No interaction between age and sex was identified (*p* > 0.10).

### 3.4. Laterality Index

The calculated laterality index (LI) had an overall mean of **0.05 ± 5.06%**, confirming negligible directional asymmetry between sides (Figure 3). Median LI was 0.0% (range: −16.7% to +12.4%). By sex, females had a mean LI of −0.18 ± 5.14%, and males +0.39 ± 4.95%, with no significant difference (*t* = −0.64, *p* = 0.52). Across age groups, mean LI values were −0.03% (<40 years), +0.86% (40–59 years), and −0.56% (≥60 years); none differed significantly (*F* (2,132) = 0.92, *p* = 0.40). Although the Shapiro–Wilk test suggested slight deviation from normality (*p* < 0.001), non-parametric tests yielded consistent results.

The mean laterality index of ±5% falls well below the 10% threshold typically considered clinically significant for vascular asymmetry, confirming that the cavernous ICA demonstrates functionally symmetric anatomy.

### 3.5. Principal Component and Cluster Analysis

Principal component analysis (PCA) was conducted to identify morphometric patterns among the three variables (left ICA, right ICA, and intercarotid distance). The first two principal components explained 89.9% of the total variance (PC1 = 56.9%, PC2 = 33.1%). PC1 represented a vessel caliber dimension, with strong loadings from the left (0.91) and right (0.92) ICA diameters. PC2 represented cranial base width, dominated by intercarotid distance (0.99). The two axes were inversely related, indicating that individuals with larger ICAs did not necessarily have wider intercarotid spacing (Figure 4).

Cluster validation using the elbow method showed optimal clustering at k = 2, with within-cluster sum of squares decreasing from 404.3 (k = 1) to 134.2 (k = 2) to 98.7 (k = 3). The average silhouette width was 0.52 for the two-cluster solution, indicating reasonable cluster separation. Bootstrap resampling (1000 iterations) yielded a Jaccard similarity coefficient of 0.81, confirming robust cluster membership.

Using the first two PCA components, a two-cluster solution was identified, labeled Morphotype 1 and Morphotype 2, comprising 61 (45%) and 74 (55%) individuals, respectively. The distribution between morphotypes does not significantly differ from an equal split (χ^2^ = 1.34, *p* = 0.25). Morphotype 1 was characterized by larger arterial diameters (mean left ICA = 5.60 mm; right = 5.62 mm) and smaller intercarotid spacing (16.7 mm), while Morphotype 2 showed smaller calibers (4.66 mm bilaterally) with wider spacing (18.0 mm). Mean age did not differ significantly between groups (52.1 vs. 49.7 years, *p* = 0.35).

Sex distribution differed significantly between morphotypes (χ^2^ = 6.38, *p* = 0.012), with males predominating in Morphotype 1 (54% male) and females in Morphotype 2 (69% female). The mean ICA diameter was significantly greater in Morphotype 1 (5.61 ± 0.28 mm) than in Morphotype 2 (4.66 ± 0.29 mm; t = 15.06, *p* < 0.001). Intercarotid distance was marginally higher in Morphotype 2 (t = −1.84, *p* = 0.068).

## 4. Discussion

### 4.1. Anatomical Context and Historical Perspective

The internal carotid artery (arteria carotis interna, ICA) is one of the most complex and clinically significant structures in neurovascular anatomy. Fisher (1938) first described the five segments of the intracranial part of the artery, and later Bouthillier et al. (1996) expanded this classification to seven segments (C1–C7), being the most widely used in neurosurgical practice [1,2,3,4,5]. Among them, the cavernous segment (C4) is of particular importance due to its location in the cavernous sinus and its close topographical relationship with the cranial nerves, venous plexuses, and pituitary region. The morphometric parameters of this segment—diameter and intercarotid distance—are of significant clinical importance in the diagnosis and surgical treatment of various diseases [2,3,4,5,6].

The internal carotid artery arises from the bifurcation of the common carotid artery approximately at the level of the fourth cervical vertebra (C4) and supplies blood to most of the cerebral hemispheres. It is divided into four main parts: cervical (C1), petrosal (C2), cavernous (C3–C4), and supraclinoid (C5–C7). The cavernous part passes through the cavernous sinus, surrounded by the sympathetic plexus and the cranial nerves (III, IV, V1, VI), with the abducens nerve being the closest. This complex anatomical configuration explains why even minimal structural variations or pathological changes can lead to clinical manifestations such as paresis of the oculomotor nerves, visual disturbances, and ischemic incidents [1,2,3,4,5,6,7,8].

### 4.2. Morphometric Findings in International Context

This study presents MRI-based morphometric data for the cavernous segment of the internal carotid artery, which correspond to those published in previous studies [1,2,3,4,5]. The mean diameter of the artery (approximately 5.0–5.3 mm) coincides with the results of Krejza et al. (2006), confirming the relative stability of this parameter across different populations [6]. The intercarotid distance (average 17.4 mm) also falls within the 15–20 mm range described by Bouthillier and Rhoton in anatomical dissections, indicating that the morphology of the skull base in the Bulgarian population does not differ significantly from global values [2,3,4,5,6]. The observed gender differences correspond to the data from the MRI study by Agarwal et al. (2020), who report slightly larger artery diameters in men [1,2,3,4,5,6,7].

Our data align closely with previously published morphometric studies across diverse populations. Baz et al. (2021) analyzed the morphometry of the internal carotid artery using CT angiography and found that the average diameter of the cavernous segment varies between 4.9 and 5.3 mm, which coincides with the results from our sample [9]. Farımaz et al. (2019) reported similar values for intervascular distance and vessel caliber and found no significant gender differences, confirming our observations of anatomical symmetry [10]. A more recent study by Cevik et al. (2024), based on three-dimensional digital subtraction angiography (3D DSA), found minimally larger diameters in men and marked variability in the angles and curves of the cavernous segment, which affect endovascular navigation [11]. Data from Dannhoff (2023), obtained using MRI TOF technique, further demonstrate significant individual variability in the configuration of the cavernous part of the artery, which may be important in planning transsphenoidal and parasellar surgical approaches [12]. Reference values for the intercarotid distance presented by Valluru et al. (2025) in a South Indian population (mean 16.1 ± 3.8 mm) are close to our results (17.4 mm) [13], confirming the similarity in basic anatomical proportions between different ethnic groups. The comparison of these data highlights the consistency of MRI and CT morphometry as a reliable method for determining normal anatomical parameters and the importance of population-oriented reference values in the interpretation of results [9,10,11,12].

### 4.3. Sex-Related Differences and Lateralization

The observed sex-related differences in ICA diameter (Cohen’s d = 0.52 for left, 0.62 for right) are consistent with known dimorphism in cranial base dimensions. The slight predominance of the difference in the right ICA diameter (*p* = 0.008) compared to the left (*p* = 0.10) may reflect subtle hemispheric asymmetries in cerebrovascular development. However, the bilateral nature of the observed trends (both sides showing larger diameters in males) suggests a systemic rather than truly lateralized mechanism, likely related to overall differences in cranial base dimensions and body size between sexes. This finding warrants further investigation with larger samples and correlation with cerebral dominance patterns and hemispheric vascular territories [9,11,14,15,16,17,18,19,20,21,22,23,24].

### 4.4. Clinical Significance of Morphotypes

The identification of two distinct morphotypes through principal component and cluster analysis has direct clinical implications for surgical and interventional planning. Morphotype 1, characterized by larger vessel caliber (mean 5.61 mm) and narrower intercarotid spacing (16.7 mm), may facilitate endovascular catheter navigation due to greater luminal diameter, but presents increased risk during transsphenoidal approaches due to reduced lateral working space. The narrower spacing increases the probability of arterial proximity to the midline surgical corridor, requiring more careful dissection and potentially limiting the extent of safe lateral exploration. Conversely, Morphotype 2, with smaller caliber (mean 4.66 mm) and wider spacing (18.0 mm), provides greater lateral margin of safety for transsphenoidal surgery but may present technical challenges for endovascular procedures in vessels below 4.5 mm diameter [4,5,6,7,8,9,10,11,12,13,16,20,21,25,26,27,28,29,30].

The sex-associated distribution of morphotypes (males predominantly Morphotype 1 with 54% prevalence, females predominantly Morphotype 2 with 69% prevalence) suggests that preoperative sex-stratified risk assessment may be warranted. Neurosurgeons performing transsphenoidal pituitary surgery should be particularly vigilant for narrow intercarotid distances in male patients, while interventionalists may anticipate more challenging catheter navigation in female patients with smaller vessel caliber. Future prospective studies should correlate these morphotypes with intraoperative findings and complication rates to validate these theoretical considerations and establish evidence-based risk stratification protocols [11,12,13,20,21,22,23,29,30,31,32,33,34,35,36].

### 4.5. Pathological Considerations

Congenital agenesis of the internal carotid artery is a rare anomaly, with a frequency of approximately 1:10,000 people. In such cases, compensatory blood supply is provided by the anterior and posterior communicating arteries, as well as by extracranial anastomoses [1,2,3,4]. In addition to congenital anomalies, the cavernous segment can be affected by acquired diseases—aneurysms, thromboses, dissections, stenoses, and carotid-cavernous fistulas [5,6,7,8,25]. Aneurysms of the cavernous and supraclinoid parts often manifest as compression of the cranial nerves due to their extradural location, but they can also cause ischemic events. Treatment includes microsurgical clipping or endovascular techniques (stent grafts, coiling, flow diverters), which achieve a high degree of safety and effectiveness [5,6,7,8,14,15,16,25,28].

A carotid-cavernous fistula is a pathological arteriovenous communication between the internal carotid artery and the cavernous sinus and may be post-traumatic or spontaneous. The Barrow classification (types A–D) is of important practical significance in the choice of therapeutic approach [10,11,12,13,29]. Thrombosis and stenosis of this segment are often the result of atherosclerosis, infections, or compression by basal tumors (meningiomas, pituitary adenomas) [20,21,22,23,29,30]. Dissections of cerebral aneurysms, although less common, carry a high risk of ischemic stroke and are often associated with genetic variations affecting the structure of the vascular wall [20,21,24,31,32,33,34,35,36,37,38,39,40,41,42]. Morphometric examination of the cavernous segment of the ICA and the intercarotid space plays an essential role in the training of neurosurgeons and in personalizing the treatment according to individual anatomy.

### 4.6. Surgical and Interventional Implications

From a clinical point of view, precise knowledge of the morphometry of the cavernous segment is essential in neurosurgical interventions in the sellar and parasellar regions. Endoscopic endonasal approaches, especially in transsphenoidal pituitary surgery, pass in close proximity to the artery. Newman et al. (2020) found that in about 10% of cases, the intercarotid distance is less than 15 mm, which significantly increases the risk of iatrogenic vascular trauma [21,24,31,32,33,34,37,38,39,41,42]. Our data confirm these observations, with 8.9% of our sample (12 of 135 patients) demonstrating intercarotid distances below 15 mm, emphasizing the need for individualized preoperative assessment using high-resolution MRI.

The results are also relevant for endovascular interventions, where the curvature and diameter of the vessel affect catheter navigation, hemodynamics, and stent placement. The determination of normative morphometric ranges in the Bulgarian population is of direct importance for optimizing the safety and effectiveness of these procedures. The morphometric characteristics of the internal carotid artery are particularly important because transsphenoidal surgical approaches pass through the posterior wall of the sphenoid sinus near the artery itself. Newman et al. found that in 10% of cases, the artery is located 4 mm from the midline in this area [34,35]. In 233 coronal magnetic resonance images, the proximity of the artery to the cavernous sinus, sphenoid sinus, and supraclinoid segment was 24.5%, 35.5%, and 39.7%, respectively, according to their measurements [21,22,30,31,32,33,34,37,38,39].

### 4.7. Biomechanical and Genetic Perspectives

Contemporary genetic and hemodynamic studies deepen the understanding of the variability of the internal carotid artery. Genomic association analyses identify loci on chromosomes 20p11 and 9p21 associated with vessel wall thickness and susceptibility to atherosclerotic changes [20,21,23,24,30,31,32,33,34,35,36,37,38,39,40]. Computer hemodynamic models (Sun et al., 2024) show that the local geometry of the vessel is crucial for blood flow and the development of aneurysms [24,37,38,39,41,42]. These morphometric parameters provide a foundation for future biomechanical modeling studies and may serve as anatomical correlates that could inform individualized disease risk assessment. The integration of morphometric, genetic, and hemodynamic data in future research will enhance understanding of the biomechanical environment of the cavernous segment and support personalized risk stratification.

### 4.8. Limitations

Several limitations should be acknowledged. First, the retrospective single-center design may limit generalizability to other populations, though our findings align well with international morphometric data from diverse ethnic groups. Second, our sample comprised patients referred for neurological evaluation of non-specific symptoms (headache, dizziness, cognitive concerns), which may introduce selection bias despite normal imaging findings. While these patients represent a neurologically asymptomatic population suitable for establishing normative values, future prospective studies in truly asymptomatic volunteers or population-based cohorts would strengthen these reference standards.

Third, measurements were performed on 1.5T MRI, which, while adequate for morphometric assessment, has lower spatial resolution than 3T systems or CT angiography. However, the excellent inter-rater reliability (ICC > 0.90) confirms measurement precision with this modality. Fourth, we assessed only static morphometry and did not evaluate dynamic parameters such as vessel tortuosity, three-dimensional curvature angles, or flow characteristics, which may provide additional clinically relevant information for endovascular planning. Fifth, the cross-sectional design precludes assessment of longitudinal changes in vascular morphology with aging or the influence of cardiovascular risk factors over time.

Sixth, we did not correlate morphometric findings with clinical outcomes, intraoperative findings, or complication rates. Prospective studies linking morphotypes to surgical complications, procedural success rates, and patient outcomes would provide valuable validation of the clinical significance we propose. Seventh, the study population does not fully reflect the ethnic diversity of modern Bulgaria or broader European populations, potentially limiting applicability to immigrant populations. Finally, while we identified two morphotypes, the clinical utility of this classification requires validation in independent cohorts and correlation with surgical difficulty scores, operative time, and complication rates.

Future studies should include prospective multicenter samples, 3T MRI or CT angiography for higher resolution, 3D rotational angiography for dynamic assessment, and computer modeling of blood flow patterns. The combination of morphometric, genetic, and hemodynamic data will allow for a better understanding of the biomechanical environment of the cavernous segment and will aid in individualized risk assessment and surgical planning.

### 4.9. Summary

This MRI-based study provides reference morphometric values specific to the Bulgarian population for the cavernous segment of the internal carotid artery. The results confirm high symmetry between the two sides, no age dependence, and moderate gender differences. Compared with international data, they confirm the reliability of MRI morphometry for in vivo assessment of vascular anatomy. These results have direct practical significance for the safe planning of transsphenoidal and endovascular interventions and contribute to a better understanding of individual anatomical variations. The identification of distinct morphotypes with sex-associated distribution patterns represents a novel contribution that may enable more personalized preoperative risk assessment.

## 5. Conclusions

This MRI-based morphometric study of 135 patients without cerebrovascular disease establishes population-specific reference values for the cavernous segment of the internal carotid artery. The mean bilateral ICA diameter was 5.09 ± 0.65 mm with an intercarotid distance of 17.4 ± 4.22 mm, demonstrating high bilateral symmetry (laterality index < 5%) and no age-related variations. Sex-related differences were identified, with males showing larger ICA diameters (Cohen’s d = 0.52–0.62), reflecting cranial base size dimorphism. Principal component analysis revealed two distinct morphotypes: Morphotype 1 (larger caliber, narrower spacing, predominantly male) and Morphotype 2 (smaller caliber, wider spacing, predominantly female), with significant implications for surgical risk stratification.

These normative values support individualized preoperative planning for transsphenoidal, endovascular, and skull-base procedures. Narrow intercarotid distances (<15 mm) were observed in 8.9% of cases, emphasizing the need for careful preoperative assessment to minimize iatrogenic vascular injury in skull base neurosurgery. The morphotype classification provides a framework for sex-stratified risk assessment, though prospective validation with clinical outcomes is required.

## Figures and Tables

**Figure 1 diagnostics-15-03072-f001:**
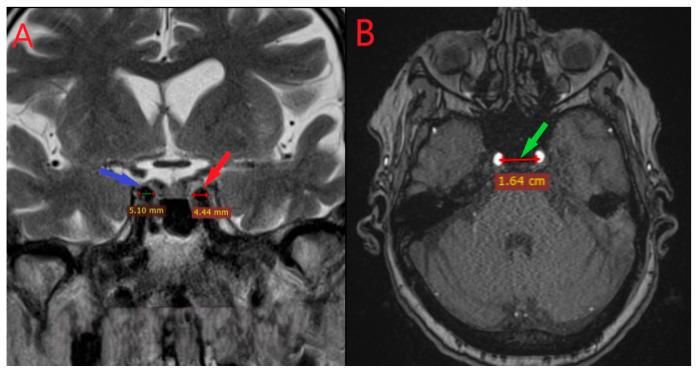
Coronal (**left panel**
**A**) and axial (**right panel B**) MRI sections showing the measurement of the diameters of the right (red pointer) and left (blue pointer) internal carotid arteries and the intercarotid distance within the cavernous segment (**panel B**, green pointer).

**Figure 2 diagnostics-15-03072-f002:**
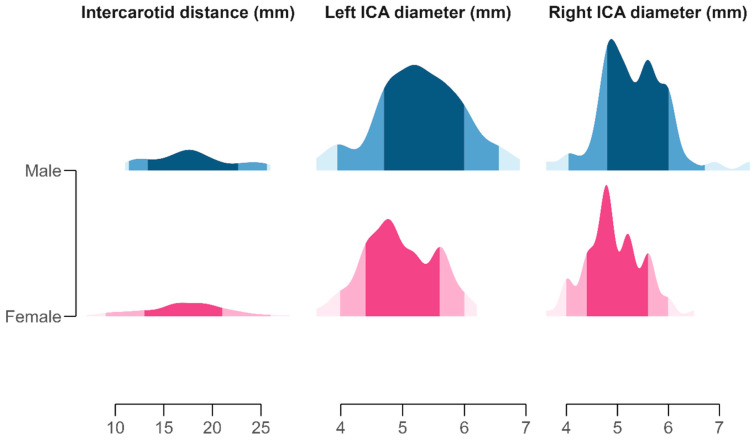
Distribution of morphometric measurements of the left and right internal carotid artery (ICA) diameters and intercarotid distance, visualized using half-eye density plots. Each slab represents the distribution of observed values, with shaded areas indicating the 50%, 80%, and 95% quantile intervals around the median. Color gradients denote sex-specific distributions: pink tones (pale rose to deep magenta) represent females, and blue tones (light cyan to navy) represent males. Facet panels correspond to individual morphometric variables and are scaled independently to highlight variation in vessel calibers and cranial spacing.

**Figure 3 diagnostics-15-03072-f003:**
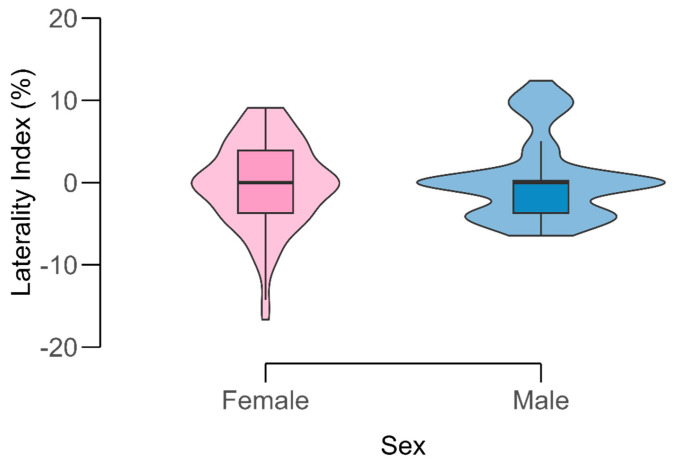
Distribution of the laterality index (LI) of the internal carotid artery (ICA) by sex. Violin plots display the full data distribution with overlaid boxplots showing median and interquartile range. Color shading distinguishes females (pink) and males (blue). The LI values are centered around zero, indicating minimal directional asymmetry between sides.

**Figure 4 diagnostics-15-03072-f004:**
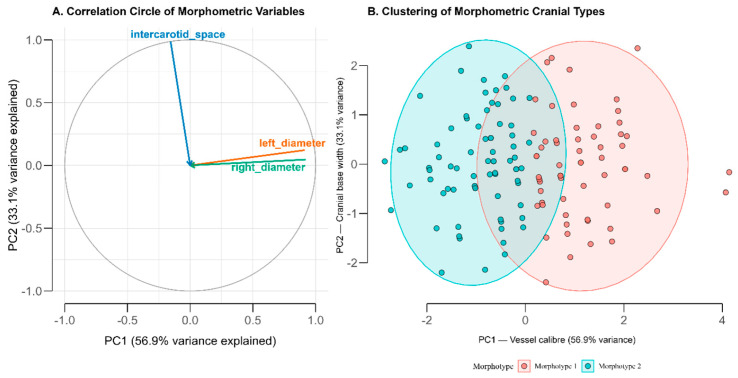
Principal Component and Cluster Analysis of Carotid Morphometric Parameters (**A**) Correlation circle showing variable loadings on the first two principal components derived from standardized measurements of the left and right internal carotid artery (ICA) diameters and intercarotid distance. The first component (PC1; 56.9% variance) reflects overall vessel caliber, while the second (PC2; 33.1%) represents cranial base width. (**B**) Two-cluster solution of individual scores in the PCA space, corresponding to distinct morphometric cranial types. Morphotype 1 (coral) is characterized by larger ICA diameters and narrower intercarotid spacing, whereas Morphotype 2 (cyan) shows smaller diameters and wider spacing. Ellipses indicate 95% confidence regions for each cluster.

## Data Availability

Anonymized morphometric data are available from the corresponding author upon reasonable request.

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
