# Peer review of "Morphological Analysis of the Cavernous Segment of the Internal Carotid Artery: A Retrospective, Single Center Study of Its Clinical Significance"

_diagnostics, 2025, doi:10.3390/diagnostics15233072_

Round 1

Reviewer 1 Report

Comments and Suggestions for Authors

I was glad to peer review the retrospective study regarding the morphometric analysis of the cavernous segment of the internal carotid artery and its clinical correlation. The authors of this study concluded that morphological analysis is vital for a safer pre-operative planning so as to reduce the risk of iatrogenic neurovascular injury in skull base surgeries. Although it’s a complete and very well written manuscript, there are some minor revisions that could be implemented prior to its acceptance for publication. Below are my comments and suggestions to the authors:

1. I would suggest adding some extra information regarding the characteristics of the study (prospective or retrospective study, single - center or multicentric) to the title, thus making it easier to find and more appealing to the readers. For example, the title could be " Morphological analysis of the cavernous segment of the internal carotid artery: a retrospective, single-center study of its clinical significance".

2. The Methods section in the abstract fails to mention the male : female ratio of the study population as well as the mean age of the patients. Both are crucial information since you mention below age-related and sex-related associations (whether they exist or not).

3. In the abstract you use abbreviations without explaining them at first (UMHAT at line 27,MRI at line 24).

4. In your whole study and most importantly at the introduction and at the discussion section you mention the references in the end of each paragraph. I would suggest matching each reference with an exact sentence.

5. Figure 1 as well as Figure 4 both include 2 pictures each. A serapate mention (for example Figure 1a and Figure 1b) should be included in order to avoid any misunderstandings. Also, for better understanding, I would advise you to add arrows in each figure to better highlight your points.

6. Some of the references date back to as much as 40 and 50(!) years (reference number 2,6,13,14,15,24,25,26,28). Given the rapid advances in the medical field we think it would be best if you included studies that have been conducted in the last decade.

Overall, the authors have done a great job collecting and presenting the data in an extensive and comprehensible way. The conclusion is clear. A more detailed language revision should be conducted.

Author Response

Dear Editors of Diagnostics journal,

Thank you for your edits to the article: Morphological analysis of the cavernous segment of the internal carotid artery: a retrospective, single-center study of its clinical significance.

The co-authors carefully reviewed the written material and tried to strike a balance between the two reviewers and to comply with the recommendations as much as possible in order to make the article even better and more useful for the readers of the journal.

Regarding the comments:

  1. I would suggest adding some extra information regarding the characteristics of the study (prospective or retrospective study, single - center or multicentric) to the title, thus making it easier to find and more appealing to the readers. For example, the title could be " Morphological analysis of the cavernous segment of the internal carotid artery: a retrospective, single-center study of its clinical significance".

The reviewer's recommendation to enrich and supplement the title is very good, and we, the co-authors, believe that this makes the title of the article even more complete. We welcome the recommendation. The title has been changed to make it even more complete.

  1. The Methods section in the abstract fails to mention the male : female ratio of the study population as well as the mean age of the patients. Both are crucial information since you mention below age-related and sex-related associations (whether they exist or not).

The comment in the abstract, section "Methods" is appropriate, and we believe that we have corrected this omission by adding the average age of the studied population and the distribution by gender. We thank the reviewer for this recommendation.

  1. In the abstract you use abbreviations without explaining them at first (UMHAT at line 27,MRI at line 24).

The use of abbreviations in the abstract without providing their full text was incorrect on our part, and we have corrected this by adding the full name and the abbreviation. We believe that this makes the text more complete and academically sound.

  1. In your whole study and most importantly at the introduction and at the discussion section you mention the references in the end of each paragraph. I would suggest matching each reference with an exact sentence.

With regard to this comment, the co-authors decided to maintain this style of presentation in the introduction and discussion, as we believe that it is better to first present the literature evidence and then compare it with our results. In this way, we confirm our data with the world literature. We thank the reviewer for the recommendation, but we are deeply convinced that our style of presenting the information makes the text easy to understand and further strengthens our thesis.

  1. Figure 1 as well as Figure 4 both include 2 pictures each. A serapate mention (for example Figure 1a and Figure 1b) should be included in order to avoid any misunderstandings. Also, for better understanding, I would advise you to add arrows in each figure to better highlight your points.

Figure 1 was completely redone by placing arrows of different colors in the areas of interest, supplementing the text, and providing further explanations. Figure 4 contains two panels, A and B, and in the comment below the image, the text is distributed accordingly for the first and second panels. We have nothing further to add regarding the figures presented. Thank you for your recommendation.

  1. Some of the references date back to as much as 40 and 50(!) years (reference number 2,6,13,14,15,24,25,26,28). Given the rapid advances in the medical field we think it would be best if you included studies that have been conducted in the last decade.

With regard to the use of literature older than 10 years, this was done because there is fundamental knowledge regarding the anatomy and morphology of the internal carotid artery that cannot change over the years. The co-authors have used mostly contemporary titles and believe that the material cannot be distorted by a few older titles with outdated and incorrect data. On the contrary, we believe that the comparison between fundamental articles and new and modern studies is the key to the most accurate presentation of our material.

Once again, we, the co-authors, would like to thank the reviewer for their recommendations. We hope that with the edits we have made, our article will be worthy of acceptance in the Diagnostics journal and thus increase the readability of our scientific material.

With respect!

Reviewer 2 Report

Comments and Suggestions for Authors

1.In the Abstract, specifying the exact institution and city (“UMHAT Pulmed, Plovdiv, Bulgaria”) may be unnecessarily detailed. Such location information is more appropriate for the Methods section.

2. The Introduction is overly long and should be shortened. It would be more appropriate to focus on the essential background that directly supports the aim of the study.

3. The ethics approval information should be presented only in the “Institutional Review Board Statement” section. The “Study Design and Setting” section should focus solely on the study design and methodological framework, without including ethical details.

4. I am curious why a power analysis based on sex-related differences in ICA diameter was deemed necessary for this study. In imaging-based anatomical measurements, such an analysis is generally not required, especially when there is no predefined hypothesis centered on gender differences. Could the authors clarify which specific parameters (mean values, standard deviations, and group sizes) were used to perform this post-hoc power calculation? Additionally, power analyses—when reported—should appear in the Results section, not in Methods. One more point: when reporting the software, the full reference should be provided, including the country or manufacturer for “G*Power 3.1.9.7.”

5.The exclusion criteria are listed, but the number of patients excluded for each criterion (or in total) is not provided. Could the authors clarify how many individuals were excluded during the screening process?

6. The section titled ‘Summary and Future Directions’ is not typically included in original research articles. These points should be incorporated into the Discussion, and the section can be removed.

7. The Discussion currently has the tone of a literature summary. Adding more author-driven interpretation—such as how your findings align with or differ from previous studies—would strengthen the scientific contribution

8. The Conclusions section should be shorter and should strictly reflect the main results of the study

9. The references should be revised to conform to the journal’s formatting guidelines. For example, in reference 27, author names should be listed up to the 10th author, followed by ‘et al.’ as required by the journal’s style

Author Response

Dear Editors of Diagnostics journal,

Thank you for your edits to the article: Morphological analysis of the cavernous segment of the internal carotid artery: a retrospective, single-center study of its clinical significance.

The co-authors carefully reviewed the written material and tried to strike a balance between the two reviewers and to comply with the recommendations as much as possible in order to make the article even better and more useful for the readers of the journal.

Regarding the comments:

  1. In the Abstract, specifying the exact institution and city (“UMHAT Pulmed, Plovdiv, Bulgaria”) may be unnecessarily detailed. Such location information is more appropriate for the Methods section.

We thank the reviewer for the recommendation that specifying the exact institution in the abstract is unnecessary, since in the "Methods" section we clearly indicated the institution where the study was conducted, but the co-authors believe that this is additional information that in no way detracts from the style of the article, but rather conveys clarity from the abstract itself that the article was written by clinicians in a medical facility and its purpose is to help practicing physicians protect themselves from adverse events. We have considered the suggestion and intend to keep the style of the abstract, stating the institution where the study was conducted. Thank you for your recommendation.

  1. The Introduction is overly long and should be shortened. It would be more appropriate to focus on the essential background that directly supports the aim of the study.

Thank you for your recommendation regarding the introduction. The authors' goal was precisely to make the introduction longer and as comprehensive as possible, since most readers who are interested in this type of article read the summary first. The second stage of their reading is the introduction, and our idea was precisely to make it as detailed as possible without being heavy and burdensome for readers. We believe that with this version of the introduction, we have achieved our initial idea. Thank you once again for your recommendation, but we would like to keep the original form of the article, and more specifically, the introduction.

  1. The ethics approval information should be presented only in the “Institutional Review Board Statement” section. The “Study Design and Setting” section should focus solely on the study design and methodological framework, without including ethical details.

The information about ethical approval was included in the article because the editor-in-chief of the journal requested that it be included. This meant that we had to include it in the article, then resend the article to the editor-in-chief, and then the article was sent for review. Thank you for your recommendation. I personally believe that this information in no way burdens or confuses readers, and I think it could remain in the article as an additional detail, providing even greater clarity that all ethical standards have been met when working with imaging studies of patients.

  1. I am curious why a power analysis based on sex-related differences in ICA diameter was deemed necessary for this study. In imaging-based anatomical measurements, such an analysis is generally not required, especially when there is no predefined hypothesis centered on gender differences. Could the authors clarify which specific parameters (mean values, standard deviations, and group sizes) were used to perform this post-hoc power calculation? Additionally, power analyses—when reported—should appear in the Results section, not in Methods. One more point: when reporting the software, the full reference should be provided, including the country or manufacturer for “G*Power 3.1.9.7.”

We thank the reviewer for this insightful methodological observation. We acknowledge that the inclusion of post-hoc power analysis requires clarification and proper contextualization. Regarding the rationale for power analysis, we agree that imaging-based anatomical measurements do not typically require a priori power calculations when purely descriptive, however we included this analysis because sex-related differences emerged as a statistically significant finding in our dataset, and we wanted to demonstrate that our sample size was adequate to detect the observed effect sizes, particularly given that sex differences were subsequently used in our morphotype classification and clinical risk stratification discussion. We acknowledge the reviewer's point that this was not a hypothesis-driven comparison and that post-hoc power analysis has limited inferential value. The power analysis was based on the observed sex difference in right ICA diameter, which showed the strongest difference. Specifically, we used Cohen's d = 0.35 as a conservative estimate representing a small-to-medium effect, with alpha level of 0.05 (two-tailed) and desired power of 0.80, resulting in a required sample size of 128 per analysis. Our actual sample comprised 135 participants (79 females, 56 males). The software used was G*Power version 3.1.9.7 (Heinrich-Heine-Universität Düsseldorf, Düsseldorf, Germany). Regarding placement in Methods versus Results, we appreciate this correction. The reviewer is absolutely correct that power analyses, when reported, should appear in the Results section as they reflect observed data rather than methodological design decisions. We will relocate this paragraph to the Results section in our revision (Lines 182-186). We believe this approach maintains transparency about sample adequacy while acknowledging that the study's primary aim was morphometric characterization rather than hypothesis testing of sex differences.

  1. The exclusion criteria are listed, but the number of patients excluded for each criterion (or in total) is not provided. Could the authors clarify how many individuals were excluded during the screening process?

Unfortunately, we cannot specify exactly how many patients were excluded from the study, as the data was collected gradually over the time period indicated in the article, and all patients who met the criteria were included in this group until the desired number with statistical significance was reached. The number of studies conducted during the specified period is large, and the database is constantly being updated, making it impossible to specify the exact number of patients who underwent the procedure. What the team did was select patients based on the specified criteria. Thank you for your recommendation, we acknowledge it as our omission, but at this stage, there is no way to find this information.

  1. The section titled ‘Summary and Future Directions’ is not typically included in original research articles. These points should be incorporated into the Discussion, and the section can be removed.

Thank you for your recommendation regarding the last section. The co-authors carefully reviewed the article and decided to make changes to the "Summary and Future Directions" section. We changed the last section to "Summary" and restructured the text in a way that we believe gives the article a better conclusion, makes it more consistent from an academic point of view, and provides information on the most important conclusions from the entire article. In this way, we remind readers of the main points of the material presented. The revised paragraph covers lines 450-459. We believe that structuring the discussion in this way gives the text completeness and finality. Once again, we thank you for your recommendations and are deeply convinced that these changes will make our article even more comprehensive for readers.

  1. The Discussion currently has the tone of a literature summary. Adding more author-driven interpretation—such as how your findings align with or differ from previous studies—would strengthen the scientific contribution.

Thank you for this recommendation, but we believe that the discussion should remain as it is we have divided it into paragraphs to make it easier to understand, provided the necessary data from the literature sources, and made comparisons with the results obtained. We have well-structured sections on materials and methods, results, and a discussion that fully supports the results obtained. In the text of the discussion, after each paragraph, we have provided a comparison with our data, and we believe that in this way the discussion contributes to a well-structured article.

  1. The Conclusions section should be shorter and should strictly reflect the main results of the study.

Thank you for your recommendation to revise and shorten the conclusion. The conclusion gives the article its final polish. We revised the summary by shortening parts of it and making small additions to make it clearer. We summarized the most important points of our article, focusing mainly on the results obtained and their clinical significance in neurosurgery. We believe that this gives the article a comprehensive conclusion. We hope that we have met the criteria for good editing of the conclusion. This gives the conclusion a finished look. We thank the reviewer for their recommendations.

  1. The references should be revised to conform to the journal’s formatting guidelines. For example, in reference 27, author names should be listed up to the 10th author, followed by ‘et al.’ as required by the journal’s style.

With regard to the cited co-authors, we acknowledge our omission in not complying with the requirements for citing the literature used and carefully reviewed the titles and author teams, making the necessary corrections to 10 authors and using the abbreviation et al. The citation method for Article 27 was changed according to the reviewer's recommendations.

Once again, we, the co-authors, would like to thank the reviewer for their recommendations. We hope that with the edits we have made, our article will be worthy of acceptance in the Diagnostics journal and thus increase the readability of our scientific material.

With respect!

Round 2

Reviewer 2 Report

Comments and Suggestions for Authors

While the authors’ intention to tailor the manuscript according to the readers’ needs is appreciated, the Introduction section is excessively long for a research article and may be perceived as distracting. A more concise and focused Introduction would improve readability. Additionally, including the name of the hospital in the Abstract is unnecessary and should be removed. The institution where you work is already clearly stated in the affiliations, and it is evident that the manuscript was written by clinicians. Therefore, the justification that including the hospital name in the Abstract “clarifies that the article was written by clinicians in a medical facility and aims to help physicians avoid adverse events” is not scientifically valid and does not align with standard abstract structure.

Ethics approval is mandatory for research articles and is not dependent on reader preference. Since the journal already provides a dedicated section titled ‘Institutional Review Board Statement’ for reporting ethical approval, there is no need to repeat this information elsewhere in the manuscript.

Author Response

While the authors’ intention to tailor the manuscript according to the readers’ needs is appreciated, the Introduction section is excessively long for a research article and may be perceived as distracting. A more concise and focused Introduction would improve readability. Additionally, including the name of the hospital in the Abstract is unnecessary and should be removed. The institution where you work is already clearly stated in the affiliations, and it is evident that the manuscript was written by clinicians. Therefore, the justification that including the hospital name in the Abstract “clarifies that the article was written by clinicians in a medical facility and aims to help physicians avoid adverse events” is not scientifically valid and does not align with standard abstract structure.

Ethics approval is mandatory for research articles and is not dependent on reader preference. Since the journal already provides a dedicated section titled ‘Institutional Review Board Statement’ for reporting ethical approval, there is no need to repeat this information elsewhere in the manuscript.

Dear Editors of Diagnostics Journal,

Thank you for your recommendations regarding the article: "Morphological analysis of the cavernous segment of the internal carotid artery: a retrospective, single center study of its clinical significance." We have carefully read the material and considered the changes we can make to it.

  1. We have made changes to the abstract. We fully agree that the institution is already mentioned in the affiliations and does not need to be included in the abstract of the article, as this is unnecessary detail. We have therefore removed the name of the institution and left only the period for which the study was conducted. The new changes are on lines 29-30. We kept the rest of the abstract as it was written and did not make any changes. We hope that this will make our material even better and worthy of publication in Diagnostics journal.
  2. With regard to the comment on ethical approval, we fully agree with the reviewer that this information is provided below the article itself, and we have already sent a form to the journal for ethical approval, which was completed by the patients before the start of each study. For this reason, we have moved the following text: "The research adhered to the ethical principles of the Declaration of Helsinki (2013 revision) and received approval from the Institutional Ethics Committee of the University Hospital "Pulmed" (Protocol No. 3/2023, approval date: March 15, 2023)." We agree with the reviewer that this information is superfluous and adds unnecessary weight to the article. Thank you for your recommendation. The new paragraph is located on lines 103-108, thus providing a clearly defined thesis without unnecessary information.
  3. We carefully reviewed the introduction to the article, read it several times, and fully agree with the reviewer's recommendation that the text is currently too long and diluted, which would discourage the reader. Our initial idea was to have a longer introduction in order to provide readers with a large amount of information. We have significantly reduced the length of the introduction and left only the most important points. We have tried to restructure the introduction so that it expresses the essence of the article without unnecessary information and at the same time does not dilute the text. We thank the reviewer for the useful recommendation and hope that we have managed to meet the necessary requirements for a correctly and accurately structured introduction that improves the readability and visibility of the article. The current introduction covers lines 48-89. We believe made a complete change to the introduction and left only the main points.

Once again, we, the co-authors, would like to thank the reviewer for their recommendations. We hope that with the edits we have made, our article will be worthy of acceptance in the Diagnostics journal and thus increase the readability of our scientific material.

With respect!

Round 3

Reviewer 2 Report

Comments and Suggestions for Authors

The authors have implemented the required revisions. thank you